# Visualizing band selective enhancement of quasiparticle lifetime in a metallic ferromagnet

Na Hyun Jo[1,2,5], Yun Wu[1,2,5], Thaís V. Trevisan [1,2], Lin-Lin Wang [1], Kyungchan Lee[1,2,3,4], Brinda Kuthanazhi[1,2], Benjamin Schrunk[1,2], S. L. Bud'ko [1,2], P. C. Canfield[1,2], P. P. Orth [1,2] & Adam Kaminski [1,2✉]

Electrons navigate more easily in a background of ordered magnetic moments than around randomly oriented ones. This fundamental quantum mechanical principle is due to their Bloch wave nature and also underlies ballistic electronic motion in a perfect crystal. As a result, a paramagnetic metal that develops ferromagnetic order often experiences a sharp drop in the resistivity. Despite the universality of this phenomenon, a direct observation of the impact of ferromagnetic order on the electronic quasiparticles in a magnetic metal is still lacking. Here we demonstrate that quasiparticles experience a significant enhancement of their lifetime in the ferromagnetic state of the low-density magnetic semimetal $EuCd_2As_2$, but this occurs only in selected bands and specific energy ranges. This is a direct consequence of the magnetically induced band splitting and the multi-orbital nature of the material. Our detailed study allows to disentangle different electronic scattering mechanisms due to non-magnetic disorder and magnon exchange. Such high momentum and energy dependence quasiparticle lifetime enhancement can lead to spin selective transport and potential spintronic applications.

[1] Division of Materials Science and Engineering, Ames Laboratory, Ames, IA 50011, USA. [2] Department of Physics and Astronomy, Iowa State University, Ames, IA 50011, USA. [3] Present address: Physikalisches Institut, Universität Würzburg, D-97074 Würzburg, Germany. [4] Present address: Würzburg-Dresden Cluster of Excellence ct.qmat, Universität Würzburg, D-97074 Würzburg, Germany. [5] These authors contributed equally: Na Hyun Jo, Yun Wu. ✉email: kaminski@ameslab.gov

The development of long-range electronic order has a strong impact on electronic structure and transport properties of quantum materials. For charge density wave (CDW) and spin density wave (SDW) orders that are characterized by a finite-$q$ wave vector, this is a result of band folding and FS reconstruction due to new (larger) periodicity in the crystal. This has been observed in a number of systems including NbSe$_2$[1], Fe pnictides[2,3], CrAuTe$_4$[4], and CeSb[5]. In contrast, for ferromagnetic (FM) order the periodicity remains unchanged, but instead the bands experience a FM exchange splitting, leading to different FSs for majority and minority carriers. The effect of such band splitting on the properties of itinerant carriers and the impact of their coupling to fluctuations of the magnetic moments are important open questions that we address here. Previous studies have focused on itinerant magnets with $3d$ transition metals, where electronic bands are broad due to the strongly correlated nature of the system and the observed effects on quasiparticle properties are small. The phenomenon of FM band splitting was demonstrated, for example, in elemental Ni[6], and a more recent angle-resolved photoemission spectroscopy (ARPES) study on Fe$_3$GeTe$_2$ also revealed FM exchange splitting with reduced quasiparticle coherence[7]. Rare-earth-based FM materials with local magnetic moments and weakly correlated conduction electron bands are a much better platform for studying the impact of FM order on electronic quasiparticle properties. Here, we investigate a FM variant of EuCd$_2$As$_2$, which has received much attention recently as a candidate magnetic Weyl semimetal[8–11] displaying FM[12,13] or antiferromagnetic order[14–16], which can be controlled by doping. These unique properties make FM-EuCd$_2$As$_2$ an ideal candidate for such studies. We show that its electronic quasiparticles experience a significant lifetime enhancement in certain bands and energy ranges. We associate this effect with the emergence of different impurity and magnetic scattering rates for majority and minority carriers due to distinct phase spaces of the respective scattering processes. This demonstrates the complexity and importance of magnetic coupling to itinerant carriers and establishes a direct connection between quasiparticle properties and transport behavior in metallic FMs.

## Results

Figure 1a, b presents the crystal structure and the temperature-dependent resistivity of EuCd$_2$As$_2$. The remaining panels Fig. 1c–j show different Fermi surface (FS) cuts and the band dispersion of EuCd$_2$As$_2$ obtained using ARPES [see panels (c–f)] and DFT calculations [see panels (g–j)]. The normalized resistivity $\rho(T)/\rho(300K)$ shows a rapid drop below $T_C$ due to loss of spin-disorder scattering. The observed upturn above the transition temperature is associated with magnetic fluctuations (see theory section below). Figure 1c displays a FS cut measured at 40 K, which is above the FM ordering temperature $T_C \approx 26$ K[12]. We observe an outer FS sheet with three-fold rotation symmetry and additional intensity at the center of the Brillouin zone. The symmetry of the outer FS sheet above $T_C$ is consistent with the three-fold symmetry of the crystal structure in $ab$ plane as shown in Fig. 1a. Small deviations from perfect three-fold symmetry noticeable in Fig. 1c, d are likely the result of matrix elements that enhances and suppresses intensity in some parts of the FS, or from the fact that the cleaved surface is not perfectly flat, which slightly distorts the emission angle of the photoelectrons during angular scans. As the sample is cooled through the FM phase transition down to 10 K, the outer trigonal FS sheet expands a little bit. What is more astonishing is that a sharp circular hole pocket emerges at the center of the Brillouin zone as shown in Fig. 1d. This is clearly the result of the FM transition, which leads to a splitting into majority and minority spin bands in this unique FM, semimetallic system.

This phenomena is better visualized in Fig. 1e, f, that show the band dispersion along the black dashed lines in (c). The orange arrows point to the hole bands crossing the Fermi level, forming the circular (inner) and trigonal (outer) FS sheets seen in panels (c, d). The green arrow points to the center hole band that sinks down rapidly to roughly 100 meV below the Fermi level as the EuCd$_2$As$_2$ sample orders ferromagnetically. The black arrow in Fig. 1e points to the possible band splitting of EuCd$_2$As$_2$ in paramagnetic state. (Details will be discussed later.) Figure 1g, h shows the DFT calculated FSs in the paramagnetic and FM states, respectively. We set $k_z = 0.17$ $(\pi/c)$, which matches relatively well with the ARPES results shown above. Note that, in order to achieve a better agreement with the ARPES results presented in Fig. 1c, d, we shifted the chemical potential in the calculations downward by roughly 300 meV for both paramagnetic and FM states. This is consistent with the presence of Eu vacancies suggested by powder X-ray data[12]. Figure 1i, j shows the band dispersion of EuCd$_2$As$_2$ calculated from DFT, showing reasonable agreement with the ARPES measurements.

In addition to the band splitting, we also observed a large enhancement of quasiparticle lifetime that accompanies the FM transition. This can be clearly seen in the raw data shown in Fig. 1e, f, where a very sharp band is present only below $T_C$. In Fig. 2a–j we show the detailed temperature evolution of the band dispersion measured using 6.79 eV photons at selected temperatures between 5 and 60 K. The results were reproduced using several samples and temperature cycling. As the sample undergoes a FM transition around $T_C = 26$ K, two significant features can be observed. One is the enhancement of the quasiparticle lifetime at the inner hole pocket, and the other is the downward shift of the fully occupied band centered at Γ with decreasing temperature below $T_C$. A key observation is that the enhancement of quasiparticle lifetime occurs predominantly for the inner hole band and only over a limited energy range, which changes with temperature. This can be seen in Fig. 2a–d, where the hole band forming the inner circular Fermi pocket is sharp until it reaches the binding energy that roughly corresponds to the top of the fully occupied center hole band (indicated by the black/red/pink/orange arrows). This effect suggests a significant interband scattering between the inner hole band and the center hole band, while intraband scattering is strongly suppressed below $T_C$. As we show in our theory modeling below, the increased lifetime of the inner band can be associated with a reduced impurity scattering rate of majority carriers (compared to minority carriers) due to phase space constraints.

To better visualize the band splitting and suppression of the scattering effect at the FM transition, we performed a detailed analysis summarized in Fig. 3. Panel (a) shows momentum distribution curves (MDCs) at the Fermi energy measured at various temperatures. The changes in the electronic structure at the magnetic transition temperature of ~26 K are quite prominent. Two well-separated bands at low temperatures merge into one broad band above the transition temperature. In addition, broad peak appears at $k_y = 0$ due to upward shift of the fully occupied center band. To demonstrate quantitatively the enhancement of quasiparticle lifetime, we plotted the full-width half maximum (FWHM) of the inner hole band extracted by using Lorentzian fitting to MDC in Fig. 3c. We also plotted the FWHM of the left side of bands, extracted from multi-peak Lorentzian fits to MDCs (shown in Fig. 3a), as a function of temperature in panel (g). The red open triangles represent the broad outer band, and the black filled squares represent the inner sharp band. Interestingly, the graph somewhat resembles the temperature-dependent resistance graph, shown in Fig. 1b, especially for the sharp band: slight upturn above the transition temperature followed by rapid decrease of FWHM below the $T_C$. Although both outer and inner

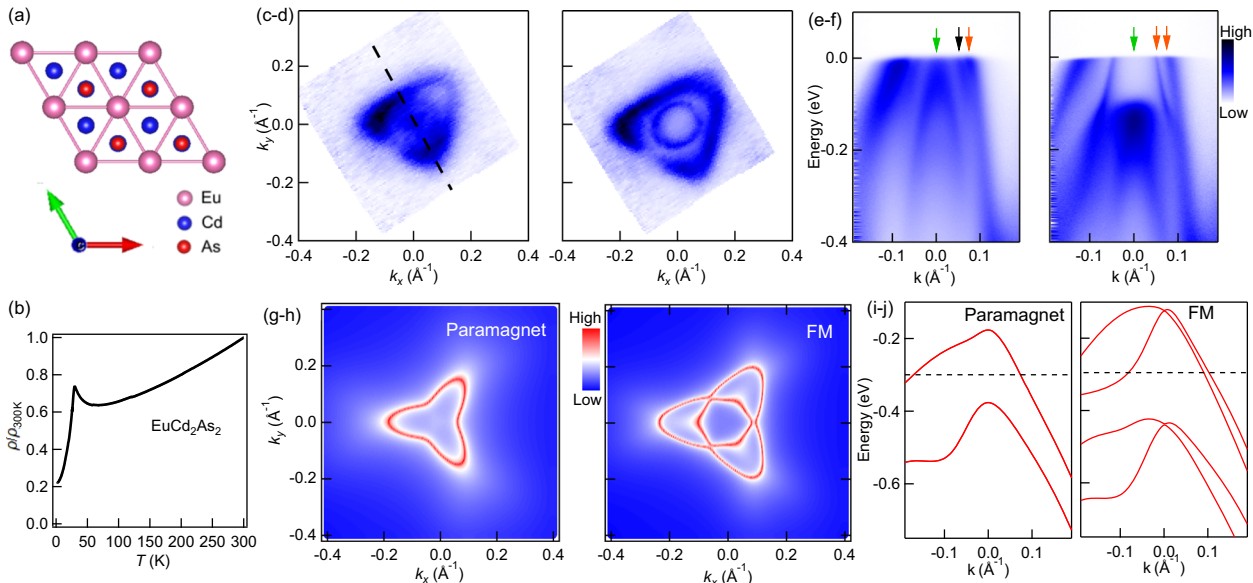

**Fig. 1 Electronic structure and Fermi surface cuts of EuCd$_2$As$_2$ from ARPES and DFT in para- and FM phase, crystal structure, and resistivity. a** Crystal structure of EuCd$_2$As$_2$. **b** Temperature-dependent normalized resistivity, $\rho(T)/\rho(300K)$, where $\rho(300K) \approx 4 \times 10^{-4}\ \omega$ cm. **c, d** FS results from ARPES measurements at 40 and 10 K, respectively, where $T_C = 26$ K is FM transition temperature. **e, f** ARPES intensity along the black dashed lines in (**c, d**) measured at 40 and 10 K, respectively. The orange arrow points to the bands crossing the Fermi level; the green arrow points to the fully occupied hole band at the center; the black arrow points to a possible splitting of the bands at 40 K. The color scale shows relative photoelectron intensity in panels (**c-f**). **g, h** FS calculated using DFT in paramagnetic and FM states, respectively. Color scale shows probability of finding available electronic state. **i, j** Band dispersion from DFT calculations along the black dashed line in (**c**). Bands undergo approximately $k$-independent rigid (Zeeman) energy shifts in the FM phase.

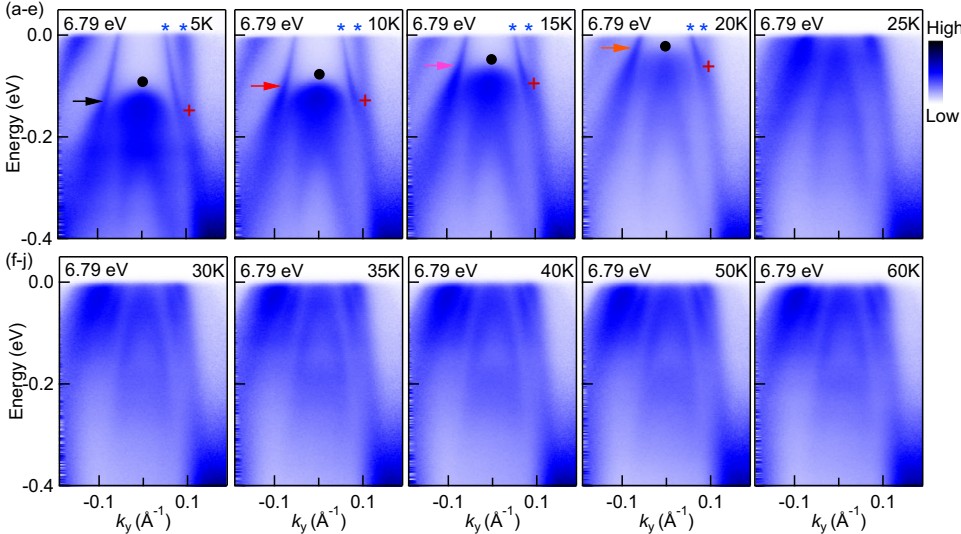

**Fig. 2 Temperature evolution of electronic band dispersion in EuCd$_2$As$_2$. a–j** Band dispersion of measured using 6.79 eV photons at indicated temperatures between 5 and 60 K. The arrows mark the energy above which enhancement of quasiparticle lifetime occurs. Black filled circles denote the top of an inner hole band, which continuously drops in energy below $T_C$. Blue stars point out the splitting of hole bands occurs on the right side. Red crosses mark the energy at which the splitting of hole bands occurs on the right side of $\Gamma$ point.

bands experience sharpening of the band, the effect is about three times larger for the inner band. If we compare the data from inner band at 5 to 30 K data, the FWHM is about six times smaller. The other interesting feature is the continuous downward shift of the center hole band with the decreasing temperature below the magnetic transition. As we can see from Fig. 2, the top of the center hole band almost touches the Fermi level and stays

at the same energy above 26 K. As the sample temperature decreases, the center hole band continuously moves downward and reaches around 150 meV below Fermi level at 5 K. In Fig. 1b, we plot the EDC at the $\Gamma$ point for several temperatures above and below $T_C$. While the EDCs measured above $T_C$ look very similar, significant changes occur upon entering FM state. The higher energy peak disappears and the main peak moves significantly to

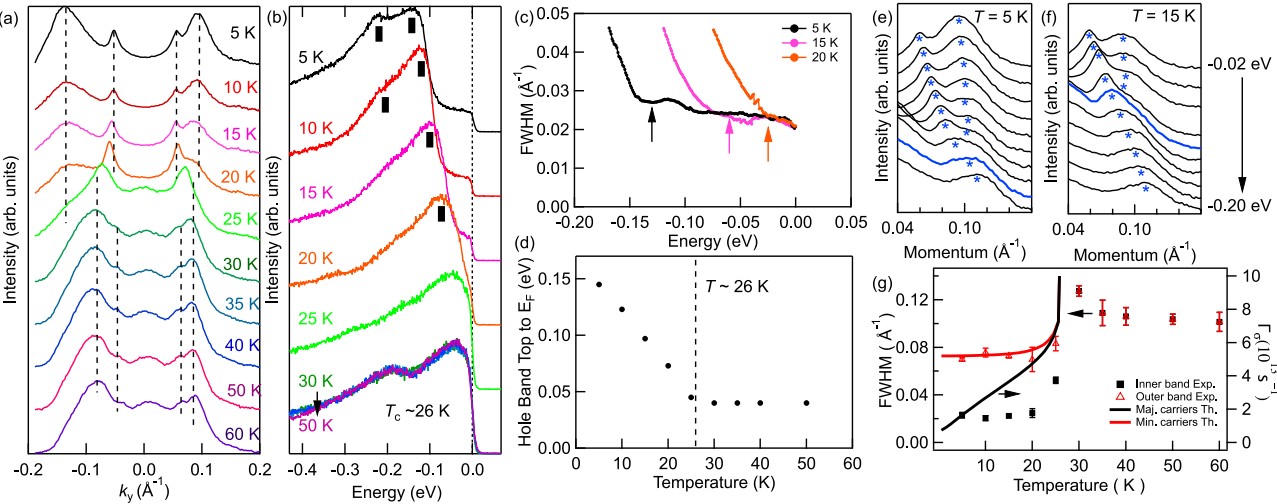

**Fig. 3 Detailed temperature evolution of the electronic structure of EuCd$_2$As$_2$ obtained using ARPES. a** MDCs at the Fermi energy measured at temperatures between 5 and 60 K. The dashed lines are a guide to eye-marking locations of MDC peaks. Comparing peak position at $T \geq 25$ K and $T = 5$ K, note that hole peak (spin) splits into two peaks in opposite directions. **b** EDCs at $k_y = 0$ at temperatures between 5 and 50 K. Black lines mark locations of peaks. Note that the central peak is (spin) split into two at low temperatures. **c** Full-width half maximum (FWHM) data obtained from Lorentzian fits to the MDCs for the inner hole band showing significant reduction of scattering (enhancement of lifetime) for energies slightly above the top of the fully occupied band, which are marked by arrows. **d** The binding energy of the fully occupied hole band top as a function of temperature. **e, f** MDCs at 5 and 15 K, respectively, for the right side of hole bands close to energies marked as blue stars in Fig. 2; −0.02 to −0.20 eV with 0.02 eV steps. Blue stars indicate the peak position. MDCs for which the two peaks merge are marked in blue. **g** FWHM of the MDC peaks on the left side in (**a**) as a function of temperature. Red triangles represent the very left peak (broad outer band), and black squares represent the second left peak (sharp inner band). Error bars represent the standard deviation of fitting the width of the peaks. The solid lines are theory results of the spin-dependent scattering rates of minority (red) and majority (black) carriers. We find good agreement under the assumption that the width in energy and in momentum space is proportional to each other.

higher binding energies. Below ~15 K, an additional shoulder appears and increases in intensity until it reaches intensity of the main peak. The energy location of the top of the center hole band seems to follow an order parameter like behavior of the magnetic ordering in EuCd$_2$As$_2$. By extracting the peak positions of the energy distribution curves, i.e., the top of the center hole band, we plotted the result in Fig. 3d showing a clear transition and enhancement of the Zeeman splitting that is a response to increasing internal field (i.e., FM order parameter) as the temperature is decreased. Figure 3e, f shows MDCs for the right side of the hole bands marked with blue stars in Fig. 2. A single band splits into two components below the transition temperature. Furthermore, the splitting point of two bands gets close to $E_F$ as temperature approaches to the transition temperature.

The enhancement of the quasiparticle lifetime below $T_C$ reflects changes in the imaginary part of self-energy. Via a Kramers–Kronig relation, it is related to the real part of the self-energy[17], which contributes to the quasiparticle binding energy. The energy shift arising from the real part of the self-energy, however, is a relatively small effect [estimated to be of the order of meV or less[18]] compared to band shifts due to the magnetic Zeeman splitting and electron-lattice interactions (e.g., due to magnetostriction), which move the central band to higher binding energies.

## Discussion

To gain insight into the experimentally observed phenomenon of band and energy-dependent quasiparticle lifetimes in the FM state, we consider a phenomenological model of itinerant holes coupled to magnetic Eu moments by exchange constant $J$. The Eu moments (spin $S = 7/2$) interact ferromagnetically with each other with exchange constant $J_{FM}$. Comparison to ARPES and DFT results (see Fig. 1), suggests using a quadratic hole-like dispersion $\xi_{\mathbf{k}} = -\hbar^2 k^2/(2m_e) - \mu$ with momentum $\mathbf{k}$, electron

mass $m_e$, and chemical potential $\mu$. We also include potential disorder that arises, for example, from Eu vacancies[12]. Importantly, all model parameters are set by comparison to experiment: $\mu$ is determined by the carrier density $n$, which we choose to be close to the value obtained from Hall measurements $n_H \simeq 2.35 \times 10^{26}$ m$^{-3}$, $JS\hbar^2 = 165$ meV follows from the observed (Zeeman) band shift in the FM state, $J_{FM}S\hbar^2 = 0.15$ meV is set by the experimental FM transition temperature $T_C = 26$ K, and the disorder strength is fixed from the observed quasiparticle lifetime $\simeq 0.04$ ps of the majority band at low temperature.

We calculate the spin-dependent electronic scattering rates in the FM phase, $\Gamma_\sigma = 1/(2\tau_\sigma)$ with lifetime $\tau_\sigma$ and $\sigma = \uparrow, \downarrow$, within the second-order Born approximation (see "Methods" and SI for details). These rates can be directly compared to the ARPES results shown in Fig. 3 [see panel (g)]. There are two scattering channels for holes in the FM phase, impurity scattering and magnetic scattering involving magnon exchange: $\Gamma_\sigma = \Gamma_{imp,\sigma} + \Gamma_{mag,\sigma}$. Both acquire an explicit spin dependence in the FM phase as we explain next. To first order in the electron-magnon coupling $J$, the hole bands split into majority (minority) bands, where the electronic spin projection is (anti-)parallel to the magnetization: $\xi_{\mathbf{k}} \to \xi_{\mathbf{k},\sigma} - \sigma \gamma(T)$ with $\gamma(T) = \frac{J\hbar^2}{2}\left[S - 0.33\left(\frac{k_B T}{J_{FM}S\hbar^2\pi}\right)^{3/2}\right]$.

The second term accounts for the reduction of saturation magnetization due to fluctuations. This results in different densities $n_\sigma$ for majority ($\sigma = \uparrow$) and minority ($\sigma = \downarrow$) carriers and different Fermi wave vectors $k_{F,\uparrow} < k_{F,\downarrow}$ for hole pockets [see Fig. 4a]. Due to this band shift the impurity scattering rates of majority and minority carriers become unequal as $\Gamma_{imp,\sigma} \propto k_{F,\sigma}$ is proportional to the density of states at the Fermi energy[19]. Majority hole carriers have a smaller FS volume and thus experience less impurity scattering, $\Gamma_{imp,\uparrow} < \Gamma_{imp,\downarrow}$, due to a reduced scattering phase space. An explicit spin dependence of the magnetic

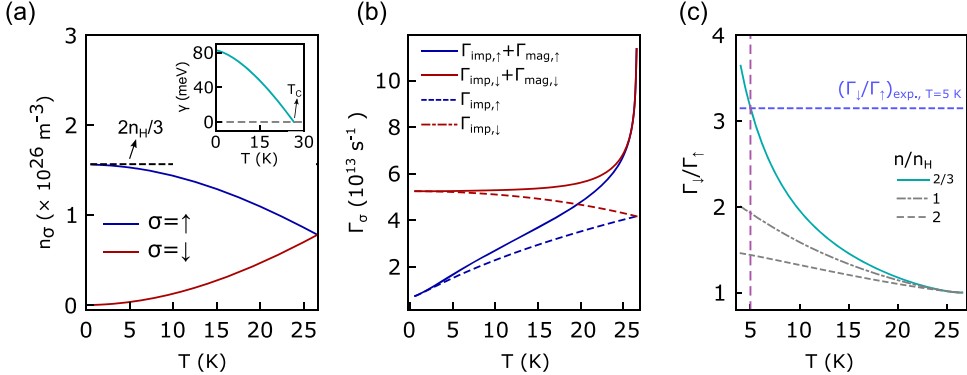

**Fig. 4 Theoretical model results. a** Spin resolved carrier (hole) density $n_\sigma$ as a function of temperature $T$ below the magnetic transition $T_C \approx 26$ K in the theoretical model with $n = n_\uparrow + n_\downarrow$. We choose $n = 2n_H/3$, where $n_H = 2.35 \times 10^{26}$ m$^{-3}$ is extracted from Hall measurements, $JS\hbar^2 = 165$ meV, and $J_{FM}S\hbar^2 = 0.15$ meV. Inset shows Zeeman energy shift $\gamma(T)$ proportional to magnetization in FM phase. **b** Total minority and majority scattering rates $\Gamma_\downarrow$ and $\Gamma_\uparrow$ (solid) as a function of $T$, where $\Gamma_\sigma = \Gamma_{\text{imp},\sigma} + \Gamma_{\text{mag},\sigma}$. Dashed lines show impurity scattering contributions, which dominate at low $T$. Upturn close to $T_C$ is caused by scattering with magnons that proliferate at the phase transition. **c** Ratio of minority over majority quasiparticle scattering rates, $\Gamma_\downarrow/\Gamma_\uparrow$, as a function of $T$ for different carrier densities $n/n_H$. The ratio increases for decreasing $T$ and $n$, and is larger than unity. The blue dashed line marks the experimental value of $\Gamma_\downarrow/\Gamma_\uparrow$ at $T = 5$ K extracted from Fig. 3g.

scattering rates, $\Gamma_{\text{mag},\sigma}$, appears to second order in $J$ and results in $\Gamma_{\text{mag},\uparrow} > \Gamma_{\text{mag},\downarrow}$ for hole-like bands[18] (see Supplementary Material).

Using parameters describing EuCd$_2$As$_2$, we find in Fig. 4b, c that the majority hole carriers experience a larger magnetic scattering rate, $\Gamma_{\text{mag},\uparrow} > \Gamma_{\text{mag},\downarrow}$, but a smaller impurity scattering rate, $\Gamma_{\text{imp},\uparrow} < \Gamma_{\text{imp},\downarrow}$. Impurity scattering dominates in the FM state (except close to $T_C$), which explains why the majority band appears sharper, $\Gamma_\uparrow < \Gamma_\downarrow$, for $T \ll T_C$. The ratio of total scattering rates, $\Gamma_\downarrow/\Gamma_\uparrow$, increases as temperature is lowered and, at fixed temperature, as $n$ becomes smaller, making the effect more pronounced in low-density semimetals such as EuCd$_2$As$_2$. Magnetic scattering only dominates close to $T_C$, but is suppressed at lower temperatures due to a reduced magnon density. The origin of the experimentally observed majority band lifetime enhancement can thus be identified with lower *impurity* scattering rates for the majority compared to minority holes. On the other hand, the sharp increase of the resistivity close to $T_C$ can be explained by scattering with magnons, which peaks at the phase transition. Finally, taking interband scattering to additional bands into account, our model naturally accounts for the experimentally observed energy dependence of the quasiparticle scattering rates, and the rapid broadening observed at energies where elastic scattering to the central hole pocket exists [see red arrow in Fig. 2a–d].

To conclude, we report a direct observation of the evolution of quasiparticle properties into the FM phase of magnetic semimetal EuCd$_2$As$_2$. In addition to energy band shifts and renormalizations proportional to the magnetization, we observe a large enhancement of the electronic carrier lifetime in selected bands and energy ranges. Investigating the temperature dependence of the lifetimes allows us to quantify the importance of different electronic scattering mechanisms, and to quantitatively relate our experimental observations to the impact of magnetic order and magnetic fluctuations on the properties of the itinerant carriers. Our work provides a direct link between quasiparticle lifetimes and loss-of-spin-disorder suppression of the resistivity, and reveals a mechanism towards spin selective transport via lifetime tuning of spin-polarized carriers.

## Methods

**Crystal growth and electrical transport**. Single crystals of EuCd$_2$As$_2$ were grown using flux growth method as described in ref. [12]. Temperature-dependent electrical transport measurement was carried out in a Quantum Design Physical Property

Measurement System (PPMS) for 1.8 K $\leq T \leq$ 300 K. The samples for electrical transport measurements were prepared by attaching four Pt wires using DuPont 4929N silver paint. The current was applied in ab plane with I = 1 mA and f = 17 Hz.

**ARPES measurements**. Samples used for ARPES measurements were cleaved in situ at 40 K under ultrahigh vacuum (UHV). The data were acquired using a tunable VUV laser ARPES system that consists of a Omicron Scienta DA30 electron analyzer, a picosecond Ti:Sapphire oscillator, and fourth harmonic generator[20]. Data were collected with photon energies of 6.05–6.79 eV. Momentum and energy resolutions were set at ~0.005 Å$^{-1}$ and 2 meV. The size of the photon beam on the sample was ~30 μm. The measurements were reproduced using several single crystals and extensive temperature cycling to exclude possibility of sample aging effects.

**DFT calculations**. Band structures with spin-orbit coupling (SOC) in density functional theory (DFT)[21,22] have been calculated using a PBE[23] exchange-correlation functional, a plane-wave basis set and projector augmented wave method[24] as implemented in VASP[25,26]. For ferromagnetic (FM) EuCd$_2$As$_2$, a Hubbard-like[27] U value of 5.0 eV is used to account for the half-filled strongly localized Eu 4f orbitals, while for non-magnetic (NM) EuCd$_2$As$_2$, the Eu 4f orbitals are treated as core electrons. For bulk band structures a Monkhorst-Pack[28] (11 × 11 × 7) k-point mesh with a Gaussian smearing of 0.05 eV including the Γ point and a kinetic energy cutoff of 318 eV have been used. Experimental lattice parameters[15] have been used with atoms fixed in their bulk positions. A tight-binding model based on maximally localized Wannier functions[29–31] was constructed to reproduce closely the bulk band structure including SOC in the range of $E_F \pm 1$ eV with Eu *sdf*, Cd *sp*, and As *p* orbitals. Then the 2D bulk band dispersions and FSs have been calculated with WannierTools[32].

**Theoretical model**. Our minimal model to describe the observed band-selective sharpening of the ARPES line-width in the FM phase of EuCd$_2$As$_2$ is composed by three parts: $H = H_c + H_{cf} + H_f$. The electronic component,

$$H_c = \sum_{\mathbf{k},\sigma} \xi_{\mathbf{k}} c^\dagger_{\mathbf{k},\sigma} c_{\mathbf{k},\sigma} + \sum_{\mathbf{k},\mathbf{q},\sigma} \sum_{\mathbf{r}_j} v_0 e^{-i(\mathbf{k}-\mathbf{q})\cdot\mathbf{r}_j} c^\dagger_{\mathbf{k},\sigma} c_{\mathbf{q},\sigma}, \quad (1)$$

accounts for electrons in a parabolic hole-like band, $\xi_{\mathbf{k}} = \hbar^2 k^2/(2m^*) + W - \mu$, with effective mass $m^* = -m_e$. Here, $m_e$ denotes the rest-electron mass and $W$ denotes the energy associated with the top of the band. The electrons experience a random potential $v(\mathbf{r}) = v_0 \sum_{j=1}^{N_{\text{imp}}} \delta(\mathbf{r} - \mathbf{r}_j)$ created by $N_{\text{imp}}$ point-like impurities at random sites $\mathbf{r}_j$. Besides,

$$H_f = -J_{\text{FM}} \sum_{\langle i,j \rangle} \mathbf{S}_i \cdot \mathbf{S}_j \quad (2)$$

describes $N$ Eu $S = 7/2$ magnetic moments on a hexagonal lattice[12] with nearest-neighbor FM coupling $J_{\text{FM}} > 0$. The interaction between the electrons and the localized moments takes the form

$$H_{cf} = -J \sum_i \mathbf{S}_i \cdot \mathbf{s}_i \quad (3)$$

where $\mathbf{s}_i = \frac{\hbar}{2N} \sum_{\mathbf{k},\mathbf{k}'} \sum_{\sigma,\sigma'} e^{-i\mathbf{R}_i\cdot(\mathbf{k}-\mathbf{k}')} c^\dagger_{\mathbf{k},\sigma} \boldsymbol{\sigma}_{\sigma\sigma'} c_{\mathbf{k}',\sigma'}$ is the spin operator of the conduction electrons, and $\mathbf{R}_i$ denotes the position of site $i$.

At low temperatures, the fluctuations about the ordered phases are small and we can map the previous Hamiltonian into an interacting electron-magnon problem via a Holstein-Primakoff transformation[33], which takes a simpler form $S_{\mathbf{q}}^z = \hbar S \sqrt{N} \delta_{\mathbf{q},0} - \hbar \sum_{\mathbf{k}} b_{\mathbf{k}}^\dagger b_{\mathbf{q}+\mathbf{k}}/\sqrt{N}$, $S_{\mathbf{q}}^+ = S_{\mathbf{q}}^x + i S_{\mathbf{q}}^y = \hbar \sqrt{2S} b_{\mathbf{q}}$, and $S_{\mathbf{q}}^- = S_{\mathbf{q}}^x - i S_{\mathbf{q}}^y = \hbar \sqrt{2S} b_{-\mathbf{q}}^\dagger$, when only a small number of magnons are excited in the system. The resulting electron-magnon interaction has four types of vertices which encode both spin-flip and spin-conserving processes (see Supplementary Information for details) and is treated perturbatively using standard diagrammatic techniques similarly as in ref. [18].

Within first-order perturbation theory, we find that the electron band experience a spin-dependent energy split $\xi_{\mathbf{k}} \to \xi_{\mathbf{k},\sigma} = \xi_{\mathbf{k}} - \sigma \gamma(T)$ due to the effective magnetic field created by the Eu moments in the ordered phase. Since the spin-up band is aligned with the magnetization direction, it is shifted downward and becomes a majority band, while the spin-down band is shifted upwards and becomes a minority band. The energy shift is temperature-dependent $\gamma(T) = \frac{J\hbar^2}{2}\left[ S - \frac{\zeta(\frac{3}{2})(k_B T)^{3/2}}{8(J_{\mathrm{FM}} \hbar^2 S n)^{3/2}} \right]$ reflecting the fact that as $T$ increases more magnons are excited in the system, which weakens the net magnetization until the FM order melts at $T = T_C$ and $\gamma(T_C) = 0$. These are also a feedback effect of the electron in the magnons, in which the lowest order effect consists of an energy shift of the magnon dispersion proportional to the difference of densities of majority and minority carriers (see Supplementary Notes for more details).

To calculate the magnetic scattering rate we apply second-order perturbation theory in the coupling $J$, which results in

$$\Gamma_{\mathrm{mag},\mathbf{k},\sigma} = \frac{\pi \mathcal{V} J^2 \hbar^3 S}{2} \int_q \left[ \sigma \, n_F(\xi_{\mathbf{k}+\sigma\mathbf{q},\bar{\sigma}}) + n_B(\Omega_{\mathbf{q}}) + \delta_{\sigma,-1} \right] \delta\left( \xi_{\mathbf{k},\sigma} + \sigma \Omega_{\mathbf{q}} - \xi_{\mathbf{k}+\sigma\mathbf{q},\bar{\sigma}} \right), \quad (4)$$

where $\bar{\sigma} = -\sigma$ and $\sigma = 1$ for spin-up. The integral in Eq. (4) was calculated numerically assuming a quadratic dispersion for the magnons $\Omega_{\mathbf{q}}$. The non-magnetic impurities were treated via self-average and the impurity scattering was calculated using the first-order Born approximation, yielding (see Supplementary Information for details)

$$\Gamma_{\mathrm{imp},\mathbf{k},\sigma} = \frac{\pi n_{\mathrm{imp}} |v_0|^2}{\hbar} \frac{1}{V} \sum_{\mathbf{q}} \delta(\xi_{\mathbf{q},\sigma} - \xi_{\mathbf{k},\sigma}), \quad (5)$$

where $n_{\mathrm{imp}} = N_{\mathrm{imp}}/V$ denotes the impurity density, where $V$ is the system volume. We set $n_{\mathrm{imp}} |v_0|^2 \approx 7.9 \times 10^{-24}\,\mathrm{eV}^2\,\mathrm{cm}^3$ in Fig. 4, which we estimated from the experimental quasiparticle scattering rate $\approx 0.25$ ps for the majority band.

## Data availability

Relevant data for the work are available[34] at the Materials Data Facility: https://doi.org/10.18126/1ume-lnp8.

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

## Acknowledgements

The authors thank R.M. Fernandes, K. O'Neal, and D.A. Yarotski for helpful discussions. This work was also supported by the Center for Advancement of Topological Semi-metals, an Energy Frontier Research Center funded by the U.S. Department of Energy Office of Science, Office of Basic Energy Sciences, Ames Laboratory under its Contract No. DE-AC02-07CH11358 (N.-H.J., T.V.T., L.L.W., B.K., P.C.C., P.P.O., A.K.). This work was also supported by the U.S. Department of Energy, Office of Basic Energy Sciences, Division of Materials Sciences and Engineering at Ames Laboratory under its Contract No. DE-AC02-07CH11358 (Y.W., K.L., B.S., S.L.B.).

## Author contributions

N.H.J. and Y.W. contributed equally to the present work. N.H.J. and B.K. grew and characterized the samples under the supervision of S.L.B. and P.C.C. Y.W., N.H.J., K.L., and B.S. acquired and analyzed ARPES data under the supervision of A.K. L.-L.W. carried out the DFT calculations. T.V.T. and P.P.O. developed the theory modeling and performed analytical calculations. All authors contributed to writing the paper.

## Competing interests

The authors declare no competing interests.
