## [Peer Review File · Nature Communications]

REVIEWER COMMENTS

Reviewer #1 (Remarks to the Author):

Referee Report for NCOMMS-21-07251-T, "Visualizing band selective enhancement of quasiparticle lifetime in a metallic ferromagnet" by N.H. Jo et al. is being considered for publication in Nature Communication.

This work aims to study the effect of ferromagnetic (FM) order on the electronic band structures near the Fermi energy (E_F), by providing direct experimental evidence on the enhancement of quasiparticle lifetime below the FM transition in EuCd_2As_2 . The authors use ARPES as the main experimental tool to directly probe the spectral function across the transition temperature (T_c). The Authors observe a monotonic increase of a valence band's binding energy (Fig.3d) and a significant decrease of momentum distribution curves (MDCs) linewidth below T_c . The Authors further interpreted the observations as due to a decrease of electronic scattering rate below T_c , which is supported by a two-band minimal model. The Authors then conclude that the calculated temperature dependent scattering rates quantitatively justify their interpretation of the data, and thus provide a direct observation of how magnetic order and magnon excitations impact the quasiparticle lifetime in FM metals. Overall, I find this study to be very interesting, and the manuscript was well written. However, some parts of the data presentation and analysis were done in a confusing way, which I hope the Authors could help clarify.

1. As the real and imaginary parts of quasiparticle self-energy are not independent of each other, the change in a band's binding energy is typically associated with a change in its lifetime. Since the Authors observed a drastic energy shift for the "fully occupied hole band" across T_c , I would expect the same band to show some change in the EDC peak width as well. Could the Authors please also provide such analysis?

I am asking this because the Authors explained the energy shift as a result of the self-energy renormalization effect from the magnon scattering. However, it seems strange to me that the Authors only observe the down-shift of the spin-up band, but not the up-shift of the spin-down band. If the energy shift is about 100meV (Fig.3d), then this band splitting should have been obvious in the data. Even if the intensity of the spin-down band is weak due to matrix element effects, the spin-up band linewidth should still show temperature dependent change across T_c .

I am wondering if it is possible that the band shift is due to extrinsic effects, such as electron doping due to air condensation on the surface in the UHV chamber?

2. In Fig.4, the Authors plot the calculated scattering rates for the 2 spin-split hole pockets as a function of temperature. However, ARPES measures band energy shifts and the MDC peak widths. I have no idea how to quantitatively compare Fig.4 with the experimental observations in Fig.3. Can the Authors please explain how to convert these quantities and thus make a direct comparison? In fact, it would be nice if the Authors could overlay the theoretical curves on the experimental datapoints in Fig.3c and 3d.

3. By comparing the band dispersion in Fig.1(e-f) to the DFT calculation, the Authors claim the samples are slightly hole-doped due to Eu vacancies. However, the estimated downward shift of chemical potential is about 450meV, which is a sizable shift even for a semimetal. It is not clear to me how the Authors reached this number, as the matching between ARPES data and DFT calculation is not great. In fact, even after the shift of the chemical potential, the ARPES data is still visibly different from the DFT result. For example, the inner hole pocket clearly shows a “kink” like feature near EF in the DFT, which is absent from the ARPES data.

Note that the DFT calculation in Fig.1(i-j) was plotted with a factor of 2 different momentum scale, compared to Fig.1(e-f). This makes the direct comparison between DFT and ARPES nonintuitive, even a bit misleading. Please plot them on the same scale.

Also, how does this shift of chemical potential translates to stoichiometry? Is this consistent with the Eu occupancy determined from the XRD refinement?

4. The FS in Fig.1d has obviously lost the $k_y=0$ mirror symmetry, and perhaps even the C_3 symmetry. The asymmetric FS is not as obvious in the 40K data. Is this a real point group symmetry breaking related to the FM transition? If not, what is the reason for the asymmetry?

5. The “rainbow” color maps in Fig. 1 and Fig. 2 make it very difficult to distinguish the fine features in the data [Nat. Commun. 11, 5444 (2020)]. In fact, I can barely see the “circular hole pocket” in Fig.1d, even though it is supposed to be sharp and clearly visible below T_c . Please consider using a more “homogeneous” color map.

6. There is only one data point between 20 to 30K that captures the big drop in the ARPES linewidth. I would really like to see more data points within this temperature range close to T_c . Also, do the Authors understand why the FWHM practically stopped changing below 15K in Fig.3c, but the “fully occupied hole band” continues to shift downward? Both effects are a result of electron-magnon scattering. I do not see why the temperature dependence should be so different.

In conclusion, the Authors presented clear experimental observations that are obviously interesting. In parallel, the Authors also presented a convincing minimal model that explains and simulates the electron-magnon interaction in an FM material. However, the link between the two is only on a qualitative level, and at times, hard to follow. For the above reasons, I would not recommend the publication of this manuscript in Nature Communications with the current format.

Reviewer #2 (Remarks to the Author):

The authors discovered band selective quasiparticle lifetime enhancement in the metallic ferromagnet EuCd₂As₂.

Using temperature-dependent ARPES measurements they found clear evidence for a ferromagnetic band splitting below

the corresponding transition temperature. The authors claim that the observed life-time enhancement in this

multi-orbital system is due to subtle band-splitting effects in the ferromagnetic phase. I fully agree with these

conclusions.

In overall the study is very timely and the spectroscopic results are state of the art. Furthermore, the authors

present in Fig.1 quite impressive calculations in the framework of relativistic density functional theory. This way,

I believe that the paper, in principal, will be of high interest for a broad scientific audience.

Although, the following theoretical interpretation, mainly in terms of many-body approaches, is somewhat exhausting.

This is not only due to the variety of parameters introduced in the different parts of the model Hamiltonian. As a

consequence it remains rather difficult to follow the quite complex argumentation for non-expert readers.

This part of the paper is quite too long and much too technical. Another point is that the suggested

many-body theory does not respect relativistic effects like spin-orbit coupling. Although, from the DFT-based

calculations it seems to follow that the selective band splitting effects are significantly influenced by spin-orbit

interaction. I found this inconsistent, and this issue needs to be discussed in more detail by the authors.

I wonder why they don't use, for example, a combination of relativistic temperature-dependent DFT+DMFT calculations

to describe the electronic and ferromagnetic degrees of freedom in terms of a unique and consistent theoretical

approach. This would reduce significantly the number of model parameters, and furthermore would help to lift the

theoretical part of paper to a level that is state of the art in this business. Last but not least I do miss

a direct comparison between the spectroscopical measurements and corresponding ARPES calculations.

In this view the present paper is a valuable work and interesting for the broad readership of Nature Communications,

but it lacks somehow in the theoretical modeling. Thus, I cannot recommend publication in Nature Communications in

the present form.

Reply to referee report of Reviewer #1:

Reviewer #1: "This work aims to study the effect of ferromagnetic (FM) order on the electronic band structures near the Fermi energy (EF), by providing direct experimental evidence on the enhancement of quasiparticle lifetime below the FM transition in EuCd₂As₂. The authors use ARPES as the main experimental tool to directly probe the spectral function across the transition temperature (T_c). The Authors observe a monotonic increase of a valence band's binding energy (Fig.3d) and a significant decrease of momentum distribution curves (MDCs) linewidth below T_c. The Authors further interpreted the observations as due to a decrease of electronic scattering rate below T_c, which is supported by a two-band minimal model. The Authors then conclude that the calculated temperature dependent scattering rates quantitatively justify their interpretation of the data, and thus provide a direct observation of how magnetic order and magnon excitations impact the quasiparticle lifetime in FM metals. Overall, I find this study to be very interesting, and the manuscript was well written. However, some parts of the data presentation and analysis were done in a confusing way, which I hope the Authors could help clarify".

We thank the referee for his/her constructive comments on our manuscript. We address them in detail in our reply and the updated manuscript.

Reviewer #1: "1. As the real and imaginary parts of quasiparticle self-energy are not independent of each other, the change in a band's binding energy is typically associated with a change in its lifetime. Since the Authors observed a drastic energy shift for the "fully occupied hole band" across T_c, I would expect the same band to show some change in the EDC peak width as well. Could the Authors please also provide such analysis?"

Reviewer #1: "I am asking this because the Authors explained the energy shift as a result of the self-energy renormalization effect from the magnon scattering. However, it seems strange to me that the Authors only observe the down-shift of the spin-up band, but not the up-shift of the spin-down band. If the energy shift is about 100meV (Fig.3d), then this band splitting should have been obvious in the data. Even if the intensity of the spin-down band is weak due to matrix element effects, the spin-up band linewidth should still show temperature dependent change across T_c."

The referee is correct in stating that the imaginary (related to lifetime) and real (one of the components of binding energy) parts of self energy are related by Kramers-Kronig

relation. Indeed, we demonstrated such relation in one of earlier papers (Phys. Rev. Lett., vol. 86, 1070–1073, (2001).). While change of the lifetime always leads to small changes in band position due to changes in real parts of the self energy, in general, the changes in the band position unrelated to real part of self energy do not have to impact the lifetime via Kramers-Kronig. This is because the real part of self energy is only one of many contributions to the binding energy and in most cases not a dominant one. Certainly, in our case, the lifetime is decreasing upon cooling, which should lead to an increase in real part of self energy and movement of the band to lower binding energies, quite opposite of what we observe. Although it is not clear the reason for this shift, most likely magnetic interaction or splitting is the prime suspect. As Referee pointed out, spin splitting is not a likely cause. On the other hand, effects such as magnetostriction can cause quite selective movement of the bands due to changes in ionic positions. We added a plot of EDC at Gamma for various temperature to better illustrate this. The new plot shows that there is possible splitting of this band, and slight sharpening of the leading edge of the peak upon cooling. We also revised this section of the manuscript to clarify those issues.

Let us now discuss the observed band shifts upon cooling below the ferromagnetic phase transition. First, as mentioned above, the dominant contribution to the energy shift of the central hole band is most likely not the Zeeman energy, but another effect, e.g., magnetostriction. This is further corroborated by our ARPES data at the lowest temperatures ($T = 5$ K and 10 K, where we can clearly observe a splitting of the inner hole band into two bands [see black rectangles in Fig. 3(b)]. We associate this splitting with a Zeeman energy shift of spin-up and spin-down bands in opposite directions. We can experimentally observe the Zeeman shift only at the lowest temperatures due to the large linewidth of the bands.

The referee is correct that according to our theory spin majority and minority bands shift in opposite directions. As shown in Fig.3(a), this is consistent with our experimental results: below $T = 25$ K (green), the hole band that crosses the Fermi energy starts to split into two peaks (clearly seen at $T = 15$ K, 10 K, and 5 K. These two bands move in opposite directions in momentum space (compare the position of the two vertical black dashed lines that trace the peaks at the lowest $T = 5$ K with the single peak at $T = 25$ K. The movement in momentum space is caused by a vertical shift of the spin split hole bands up and down in energy, respectively.

Finally, the linewidth (= inverse lifetime) of the outer (spin-down) hole band, $\Gamma_{\downarrow} =$

$1/(2\tau_{\downarrow})$ should indeed be temperature dependent according to our theoretical analysis [see Fig. 4(b)]. Theoretically, we find that the linewidth Γ_{\downarrow} decreases as temperature is lowered, just less than the inner hole band, cf. red and blue curves in Fig. 4(b). The change of the linewidth of the broader outer band, however, is more challenging to extract, since the band is in general much broader than the inner hole band. Therefore, we find that the sharpening of the inner hold band is indeed much more clearly observed in ARPES.

Reviewer #1: "I am wondering if it is possible that the band shift is due to extrinsic effects, such as electron doping due to air condensation on the surface in the UHV chamber?"

We understand the referee's concern. As we had the same worries, we checked the reproducibility upon multiple temperature cycling and using multiple crystals. To be more specific, we cleaved the sample when it reached thermal equilibrium at the base temperature, and then measured the temperature dependence data upon warming. After we obtained the high temperature data, we cooled down again and reproduced low temperature data. Therefore, we do not believe that our results are due to air condensation on the surface in the UHV chamber. We added comments to that extend in the revised version.

2. In Fig.4, the Authors plot the calculated scattering rates for the 2 spin-split hole pockets as a function of temperature. However, ARPES measures band energy shifts and the MDC peak widths. I have no idea how to quantitatively compare Fig.4 with the experimental observations in Fig.3. Can the Authors please explain how to convert these quantities and thus make a direct comparison? In fact, it would be nice if the Authors could overlay the theoretical curves on the experimental data points in Fig.3c and 3d.

The theoretical results indeed can be directly related to the experiment. First, we associate the experimentally observed band energy shifts with the Zeeman energy shift in the FM phase between spin-up (majority) and spin-down (minority) carrier bands. Theoretically, this Zeeman shift is given by

$$\gamma(T) = \frac{J\hbar^2}{2} \left[S - \frac{\zeta(\frac{3}{2})(k_B T)^{3/2}}{8(J_{FM} S \hbar^2 \pi)^{3/2}} \right],$$

with parameters J, J_{FM}, S given in the manuscript. To highlight that connection, we have updated the inset of Fig. 4(a) to show the band energy difference as a function of temperature T . Within the assumption that the experimentally observed shift is dominated by the Zeeman shift, the quantity $\gamma(T)$ corresponds exactly to the experimentally observed energy

shift between the two outer hole bands, as shown in Fig. 1(e) and Fig. 3(e). In fact, we use the experimentally observed energy difference at $T = 5$ K, $\gamma(T = 5K) = 150\text{meV}$, to set the value of the coupling constant J describing the coupling between local Eu moments and the itinerant carriers. Using this value of $J = 150\text{meV}/S$ (where $S = 7/2$ for Eu moments), our theoretical calculation thus reproduces the experimentally observed energy shift at $T = 5$ K.

Second, the theoretically calculated linewidths $\Gamma_\sigma = 1/(2\tau_\sigma)$ can also be directly compared to experiment. Here, τ_σ denotes the carrier's lifetime and $\sigma = \uparrow, \downarrow$. It is worth emphasizing that the theoretically obtained linewidths Γ_σ , which are shown in Fig. 4(b) (solid lines), show a strikingly similar temperature dependence as the experimentally observed linewidths, which are depicted in Fig. 3(g).

In both theory and experiment, the linewidths of the two bands (which merge above T_c) are identical for $T > T_c$ and show a maximum at T_c . From our theoretical modeling, this behavior can be understood as an enhancement of magnon-exchange (magnetic) scattering close to the magnetic phase transition. For temperatures $T < T_c$, the key experimental observation is that the linewidth is smaller for the inner hole band than for the outer one [compare black squares to red triangles in Fig. 3(g)]. This exact behavior is also borne out of our theoretical calculation, where we find that the spin-up (majority) linewidth (of the inner band) is significantly smaller than the spin-down (minority) linewidth (of the outer band) [see Fig. 4(b)]. We can therefore understand the experimental observation as arising from the reduced impurity scattering rate of the majority (inner) hole band compared to the minority (outer) band, which is due to the different values of $k_{F,\sigma}$.

3. By comparing the band dispersion in Fig.1(e-f) to the DFT calculation, the Authors claim the samples are slightly hole-doped due to Eu vacancies. However, the estimated downward shift of chemical potential is about 450meV, which is a sizable shift even for a semimetal. It is not clear to me how the Authors reached this number, as the matching between ARPES data and DFT calculation is not great. In fact, even after the shift of the chemical potential, the ARPES data is still visibly different from the DFT result. For example, the inner hole pocket clearly shows a “kink” like feature near EF in the DFT, which is absent from the ARPES data.

The 450 meV was a typo and the shift of E_F is 300 meV. The size of E_F shift in Fig.1(j) for FM DFT band structure is to match the top of the inner hole at 10K to be at EF-100

meV in Fig.1(f) from ARPES. Then the EF is shifted accordingly with the same size in Fig.1(g-i). The ARPES measurements are across the FM transition, while DFT calculations are only at 0 K. Although certain detail, such as the kink-like feature of the inner hole band, does not have a great match, the overall profiles of the FS in Fig.1(c) and (d) are compared well with (g) and (h) and the main feature of the band splitting and the downward shift of the inner hole upon the FM transition have been captured by the DFT calculations.

4. Note that the DFT calculation in Fig.1(i-j) was plotted with a factor of 2 different momentum scale, compared to Fig.1(e-f). This makes the direct comparison between DFT and ARPES nonintuitive, even a bit misleading. Please plot them on the same scale.

Thank you for pointing this out. We changed the scale for Fig. 1 (i-j) the same as Fig. 1 (e-f).

5. Also, how does this shift of chemical potential translates to stoichiometry? Is this consistent with the Eu occupancy determined from the XRD refinement?

From the integrated density of states in FM DFT calculation and assuming a rigid band shift, the 300 meV shift of EF corresponds to a deficiency of 0.031 electron or 0.015 Eu²⁺ vacancy per formula unit. From the high-energy X-ray measurements for this sample “FM(slat)” as listed in the supplementary Table II in Ref.12, the Eu site occupancy is 0.989 or 0.011 Eu vacancy. Thus, the size of EF shift estimated from ARPES and DFT calculation is in line with the Eu site occupancy from the XRD refinement.

6. The FS in Fig.1d has obviously lost the $k_y=0$ mirror symmetry, and perhaps even the C3 symmetry. The asymmetric FS is not as obvious in the 40K data. Is this a real point group symmetry breaking related to the FM transition? If not, what is the reason for the asymmetry?

Thank you for pointing this out. We do not believe that the small distortions visible in the shape of FS at low temperatures are intrinsic. Matrix elements that can certainly result in change of weight of the bands as well as small curvature of cleaved surfaces can move the location of FS a little when scanning k_x momentum (this is done by mechanically rotating the sample). While important, investigation of these possibilities is beyond the scope of this work as they do not affect key conclusions. We added comment addressing this issue.

7. The “rainbow” color maps in Fig. 1 and Fig. 2 make it very difficult to distinguish

the fine features in the data [Nat. Commun. 11, 5444 (2020)]. In fact, I can barely see the “circular hole pocket” in Fig.1d, even though it is supposed to be sharp and clearly visible below T_c . Please consider using a more “homogeneous” color map.

Thank you for pointing this out. We changed to a more homogeneous color map.

8. There is only one data point between 20 to 30K that captures the big drop in the ARPES linewidth. I would really like to see more data points within this temperature range close to T_c .

We agree with the Referee that it would be nice to have a higher density of data points, especially in this region. However, the temperature steps of 5 K are already quite small, we do not think that measuring in-between points would alter any of the main conclusions of the manuscript. While the linewidth sharply drops below the magnetic phase transition, it levels off towards lower temperatures. This is also observed in our theoretical calculation and can be understood as the proliferation of magnons close the magnetic phase transition, where a large enhancement of magnetic fluctuations occur. A sharp decrease of the linewidth just below the phase transition is therefore not unexpected.

As we discussed in point 1, the movement of the fully occupied band is likely dominated by magnetostriction or another contribution, but not due to the Zeeman energy shift. The Zeeman shift occurs on top of the main shift, as alluded to above and shown by the separation of peaks at low T, indicated by the black squares in Fig. 3(b). The saturation of FWHM below 15K can have several possible causes. Although our combined momentum/energy instrumental resolution is $\sim 0.01\text{\AA}^{-1}$, the actual measured FWHM will be affected by factors such as the flatness of the sample surface. Such a contribution would be consistent with the topic discussed in point 4. There is of course the possibility that this is intrinsic if quasiparticles in this band are more sensitive to impurities. We are not able at this time to distinguish between intrinsic and extrinsic effects responsible for this saturation. We commented on this in the revised manuscript.

7. Also, do the Authors understand why the FWHM practically stopped changing below 15K in Fig.3c, but the “fully occupied hole band” continues to shift downward? Both effects are a result of electron-magnon scattering. I do not see why the temperature dependence should be so different.

The saturation of the linewidth at lower temperatures is also observed in our theoretical

analysis. There, it originates from the fact that impurity scattering dominates at lower temperatures, which is only weakly temperature dependent. This can explain the experimentally observed saturation of the linewidths of the two outer bands at lower temperatures. Note that the impurity scattering rate is different for spin-up and spin-down bands because of the different density of states, which again agrees well with our experimental observations. In contrast, the energy shift of the central hole band is directly proportional to the magnetization in the sample and therefore continues to decrease towards lower temperatures.

8. In conclusion, the Authors presented clear experimental observations that are obviously interesting. In parallel, the Authors also presented a convincing minimal model that explains and simulates the electron-magnon interaction in an FM material. However, the link between the two is only on a qualitative level, and at times, hard to follow. For the above reasons, I would not recommend the publication of this manuscript in Nature Communications with the current format.

We appreciate the Referee's time and thorough comments on the manuscript. We highlighted the direct link between our experimental and theoretical results in the modified manuscript by updating the inset of Fig.4 and adding clarifying statements in the text.

Reply to referee report of Reviewer #2:

The authors discovered band selective quasiparticle lifetime enhancement in the metallic ferromagnet EuCd₂As₂. Using temperature-dependent ARPES measurements they found clear evidence for a ferromagnetic band splitting below the corresponding transition temperature. The authors claim that the observed life-time enhancement in this multi-orbital system is due to subtle band-splitting effects in the ferromagnetic phase. I fully agree with these conclusions.

In overall the study is very timely and the spectroscopic results are state of the art. Furthermore, the authors present in Fig.1 quite impressive calculations in the framework of relativistic density functional theory. This way, I believe that the paper, in principal, will be of high interest for a broad scientific audience.

We thank the referee for the careful reading of our manuscript and the overall positive assessment.

1. Although, the following theoretical interpretation, mainly in terms of many-body approaches, is somewhat exhausting. This is not only due to the variety of parameters introduced in the different parts of the model Hamiltonian. As a consequence it remains rather difficult to follow the quite complex argumentation for non-expert readers. This part of the paper is quite too long and much too technical.

We appreciate the referee's criticism and have simplified the theory discussion in the manuscript. We have also included a table with the different model parameters and how they are determined from the experimental observations. Finally, we have delegated some of the more technical aspects in the Methods section of the paper for the interested experts and focussed the discussion in the main text on the physical insight that can be drawn from the theory. We hope that the Referee will find the revised version significantly improved.

2. Another point is that the suggested many-body theory does not respect relativistic effects like spin-orbit coupling. Although, from the DFT-based calculations it seems to follow that the selective band splitting effects are significantly influenced by spin-orbit interaction. I found this inconsistent, and this issue needs to be discussed in more detail by the authors. We politely disagree with this interpretation of the DFT results. As shown in the (new) Fig. 1(g), spin-orbit coupling effects on the band splitting are actually rather small. The band

structure in the magnetically ordered phase experiences an almost k -independent rigid band shift. We have highlighted this aspect in the modified figure caption. More importantly, such a rigid, k -independent band shift is also observed in our ARPES experiments. Therefore, we disagree that effects of spin-orbit coupling on the band splitting in the ferromagnetic phase are an absolutely important ingredient that must be taken into account in our many-body model.

We note that spin-orbit coupling could certainly be included straightforwardly in our model by adding it to the bare band structure. However, this would add an unknown model parameters to the theory, which we would like to avoid. Note that currently the numerical values of all model parameters $\{n_0, J, J_{\text{FM}}, n_{\text{imp}}|v_0|^2, S = 7/2\}$ are directly extracted from comparison to experimental ARPES data, with the exception of the electron effective mass m_e^* , which we set to the bare electron mass for simplicity. This simplification does not affect our conclusions.

3. I wonder why they don't use, for example, a combination of relativistic temperature-dependent DFT+DMFT calculations to describe the electronic and ferromagnetic degrees of freedom in terms of a unique and consistent theoretical approach. This would reduce significantly the number of model parameters, and furthermore would help to lift the theoretical part of paper to a level that is state of the art in this business.

We think that a phenomenological many-body model whose parameters are directly extracted from experimental data is the preferred approach to understand and interpret our experimental results. This is because it allows us to directly associate different experimental observations with an underlying physical mechanism. For example, our many-body model gives the important insight that the different linewidths of the inner and outer hole bands arise from a different impurity scattering lifetime in the system (and not from magnon-exchange scattering as one may have naively assumed). Based on this, we predict that the ratio of minority over majority scattering rates, $\Gamma_{\downarrow}/\Gamma_{\uparrow}$, increases with decreasing carrier density [see Fig. 4(c)], which can guide further experimental work.

Furthermore, results from our theoretical model analysis explain the sharp upturn of the linewidth close to the ferromagnetic phase transition as the effect of magnon-exchange magnetic scattering, which dominates over impurity scattering in this regimes $T \lesssim T_c$ [see Fig. 4(b)]. Finally, with all model parameters fixed by experimental observations, our the-

oretical curves give an excellent qualitative agreement with the temperature dependent linewidth of both inner and outer hole bands (see updated Fig. 3(g) and our answer to the next question).

Regarding our methodological approach: the diagrammatic calculation we perform and describe in detail in the Supplementary Information is a well-established state-of-the-art approach to calculate the lifetime of quasiparticles from scattering with impurities and magnons in the system.

Last but not least I do miss a direct comparison between the spectroscopical measurements and corresponding ARPES calculations.

We included such a direct comparison in the updated manuscript in Fig. 3(g), where we find excellent agreement for parameters that are extracted directly from our experimental ARPES data.

In this view the present paper is a valuable work and interesting for the broad readership of Nature Communications, but it lacks somehow in the theoretical modeling. Thus, I cannot recommend publication in Nature Communications in the present form.

We have significantly modified the manuscript to improve the theoretical modeling part and hope the paper is ready for publication now.

REVIEWERS' COMMENTS

Reviewer #1 (Remarks to the Author):

Referee Report for NCOMMS-21-07251-T revision 1, "Visualizing band selective enhancement of quasiparticle lifetime in a metallic ferromagnet" by N.H. Jo et al., is being considered for publication in Nature Communication.

The present manuscript is an interesting and detailed work studying the quasiparticle band renormalization across the ferromagnetic transition. The Authors combine ARPES, DFT and perturbation theory to study the electronic band structures in EuCd_2As_2 at various temperatures. Based on the direct observation of the rigid shifting and significant sharpening for selected bands, the Authors provide insights into the evolution of quasiparticle lifetimes across the ferromagnetic transition. Through theoretical investigations, the Authors further provide quantitative explanations to the data. I believe this work help elucidate the various scattering mechanisms in magnetic semimetals, and thus highly interesting to a broad community of researchers working on spintronics and quantum materials. Therefore, I would recommend the publication of this manuscript in Nature Communication.

The Authors have answered all my questions and concerns in the reply. I have just one minor suggestion. In Fig.3g, the experimental data points have the unit of $1/A$, whereas the theoretical curves are scattering rate and thus have the unit of frequency (proportional to energy). I would suggest adding a right-axis with proper units, and also include a sentence in the main text to explain how the two are related.

There is a small typo (an extra equal sign) in Fig.4 caption line 3 before "is extracted from Hall measurements".

Reviewer #2 (Remarks to the Author):

With their response letter the authors were able to convince me that the paper is indeed a highly innovative work which deserves publication in a high ranking journal like Nature Communications. The changes made in the manuscript significantly improve the readability. I believe that the revised manuscript will receive a lot of attention not only in a specialized community. Furthermore the current version of the manuscript has successfully

addressed all technical questions and this way presents well refined and coherent results. I recommend the manuscript for publication in Nature Communications in its current form with no further modifications necessary.

RESPONSE TO REVIEWERS' COMMENTS IN BLUE

Reviewer #1 (Remarks to the Author):

Referee Report for NCOMMS-21-07251-T revision 1, “Visualizing band selective enhancement of quasiparticle lifetime in a metallic ferromagnet” by N.H. Jo et al., is being considered for publication in Nature Communication.

The present manuscript is an interesting and detailed work studying the quasiparticle band renormalization across the ferromagnetic transition. The Authors combine ARPES, DFT and perturbation theory to study the electronic band structures in EuCd_2As_2 at various temperatures. Based on the direct observation of the rigid shifting and significant sharpening for selected bands, the Authors provide insights into the evolution of quasiparticle lifetimes across the ferromagnetic transition. Through theoretical investigations, the Authors further provide quantitative explanations to the data. I believe this work help elucidate the various scattering mechanisms in magnetic semimetals, and thus highly interesting to a broad community of researchers working on spintronics and quantum materials. Therefore, I would recommend the publication of this manuscript in Nature Communication.

We would like to thank the Referee for careful reading of the manuscript all constructive comments and recommending publication.

The Authors have answered all my questions and concerns in the reply. I have just one minor suggestion. In Fig.3g, the experimental data points have the unit of $1/\text{\AA}$, whereas the theoretical curves are scattering rate and thus have the unit of frequency (proportional to energy). I would suggest adding a right-axis with proper units, and also include a sentence in the main text to explain how the two are related.

Thank you for sugegsting this. We now added scaled axis on right side and explained the conversion between two quantities.

There is a small typo (an extra equal sign) in Fig.4 caption line 3 before “is extracted from Hall measurements”.

Thank you for pointing this out - we now fixed this typo.

Reviewer #2 (Remarks to the Author):

With their response letter the authors were able to convince me that the paper is indeed a highly innovative work which deserves publication in a high ranking journal like Nature Communications. The changes made in the manuscript significantly improve the readability. I believe that the revised manuscript will receive a lot of attention not only in a specialized community. Furthermore the current version of the manuscript has successfully

addressed all technical questions and this way presents well refined and coherent results. I recommend the manuscript for publication in Nature Communications in its current form with no further modifications necessary.

We would like to thank the Referee for careful reading of the manuscript all constructive comments and recommending publication.